# 2D-LiDAR-Sensor-Based Retaining Wall Displacement Measurement System

**Jun-Sang Kim, Gil-yong Lee and Young Suk Kim \***

Department of Architectural Engineering, Inha University, Incheon 22212, Korea
\* Correspondence: youngsuk@inha.ac.kr

**Abstract:** The displacement of retaining walls is measured using inclinometers in order to evaluate the safety of the wall. However, inclinometers have three problems: they (1) are difficult to install, (2) have local displacement detection, and (3) are measured using manpower. Consequently, a two-dimensional (2D) LiDAR sensor-based retaining wall displacement measurement system that facilitates installation and three-dimensional (3D) displacement detection (more economically feasible than inclinometers) was developed in order to overcome the aforementioned limitations. The developed system collects 3D point cloud data about the retaining wall by rotating the 2D LiDAR sensor 360° at a constant speed. Laboratory experiments were performed using a simulated deformation model to evaluate the displacement measurement performance of the system, which had a root-mean-square error of 2.82 mm at approximately 20 m. The economic feasibility of the system was analyzed, which revealed that the system was economically feasible, with a benefit/cost ratio and breakeven point of 3.52 and 2.71 years, respectively.

**Keywords:** retaining wall; displacement measurement; stability; measurement system; economic feasibility





## 1. Introduction

During underground excavation at a construction site, retaining walls are constructed in order to prevent adjacent ground from collapsing [1]. Since the collapse of the constructed retaining wall considerably impacts the community and causes damages regardless of the construction scale, appropriate preventive measures are required on construction sites. Also, because the deformation in retaining walls generally increases before its collapse, it must be continuously managed to prevent any collapse [2,3]. Measuring instruments, such as inclinometers, can determine the displacement of retaining walls, which cannot be done with the natural eyes. Inclinometers are generally used to determine the displacement of retaining walls and are among the main measuring instruments used in evaluating the stability of retaining walls. However, inclinometers have the following problems: (1) the inclinometer casing must be installed near the retaining wall in advance using boring equipment; (2) selecting a cross-section of the retaining wall where large deformation occurs is important. This can be achieved by consulting measurement experts and installing multiple inclinometers at the cross-section, since these experts can only identify the displacement of the local two-dimensional (2D) cross-section of the retaining wall [4]; (3) acquiring considerable human resources can be challenging, since measurement experts collect, analyze, and interpret measurement data only on construction sites depending on the measurement cycle [5].

To address the problems inclinometers have, several studies and techniques have been conducted and developed, respectively. These techniques are divided into contact (which use conventional inclinometers) [6–11] and non-contact measurement methods (which use the latest technologies such as laser and image sensors) [1–5,12–17] depending on the measurement method. On the one hand, contact measurement methods reduce the frequency at which measurement experts visit construction sites, since they can automatically

collect measurement data using wireless network technologies and evaluate the stability of the retaining wall using said data. However, using wireless network technologies is more expensive than conventional inclinometers. Additionally, installing and selecting inclinometers and optimal measurement positions, respectively, is difficult. On the other hand, non-contact measurement methods determine three-dimensional (3D) displacement in different areas, such as tunnels [18], dams [19], landslides [20], and bridges [21]. This is because they are easier to install compared to contact measurement methods and can measure the overall displacement of the retaining wall. However, regarding terrestrial, laser scanner (TLS)-based non-contact measurement methods, the scanner is expensive, and the measuring personnel must install and dismantle the equipment for each measurement, because it cannot be installed permanently outdoors. Vision-based displacement measurement is less expensive than TLS. Therefore, a target plate is required to increase the accuracy of the deformation analysis results. This method has limited applicability in inclement weather such as rain and snow. Consequently, the objective of this study is to develop a retaining wall displacement measurement system that can resolve inclinometer limitations and previously developed methods for retaining walls that were constructed during underground excavation at construction sites. In order to evaluate the performance and applicability of the developed system, its displacement measurement performance and economic feasibility were analyzed. If the developed system is commercialized and used for measuring retaining walls, it is expected that retaining wall displacement management will be cost-effectively performed vis-a-vis the conventional measurement method.

## 2. Retaining Wall Displacement Measurement Review

### 2.1. Problems with Retaining Wall Displacement Measuring Instruments

The stability of retaining walls is assessed complexly. This is done by examining the water level, ground subsidence, and the displacement of the retaining wall using different measuring instruments, such as inclinometers, water-level meters, strain gauges, load cells, and tiltmeters. The lateral displacement of the retaining walls is an important factor for identifying the behavior of the retaining wall [4,12]. Consequently, in several countries, including the United States, China, and Korea, the lateral displacement threshold is set, while carefully managing the lateral displacement of the retaining wall [22–24].

Generally, inclinometers are used at construction sites to determine the lateral displacement of retaining walls. They are installed before underground excavation to examine the lateral displacement of the retaining wall during the excavation stage and the retaining wall retention period. To measure the lateral displacement of the retaining wall using an inclinometer, a hole with a depth above the excavation depth of 2–3 m is bored at a specific distance from the retaining wall, and the inclinometer casing is installed in the hole. This is followed by grouting in the inclination casing and the installation of a protective cover, as shown in Figure 1. After the inclinometer casing is installed, a worker inserts an inclinometer into the casing to measure the lateral displacement of the retaining wall according to the measurement cycle. However, measuring the lateral displacement of the retaining wall using an inclinometer has three problems:

1.   Difficulty installing inclinometer casing

Since the inclinometer casing must be installed around the retaining wall after boring to measure the displacement of the retaining wall using an inclinometer [5,25], heavy equipment (boring equipment) and manpower are required. Additionally, if the inclinometer casing is twisted while being inserted into the bored hole, the inclinometer will get stuck, which will cause errors when measuring the retaining wall displacement.

2.   Measuring 2D local lateral displacement of a retaining wall

Owing to the nature of inclinometers wherein only the 2D cross-section of the retaining wall can be measured [5,25,26], it is quite challenging to determine the overall lateral displacement of the retaining wall. Further, it is difficult for measurement experts to identify areas where maximum deformation may occur and select the best optimal inclinometer

installation position based on the results of the ground survey at the construction site and the design of the retaining wall [1].

3. Measurement by manpower

The lateral displacement data of the retaining wall are collected at each measurement point after the inclinometer is moved to measurement depth when a measurement expert visits the construction site once or twice a week (measurement cycle). However, as this measurement method that is based on manpower, errors may be detected in the collected data, since the measurement point changes depending on the expertise of the measurement expert [27]. Moreover, considerable manpower is required because measurements must be done at all measurement points near the retaining wall. Additionally, since it takes approximately one week for the measurement expert to prepare an analysis report using the collected data, it is difficult for a construction site manager to evaluate the stability of the retaining wall in real time.

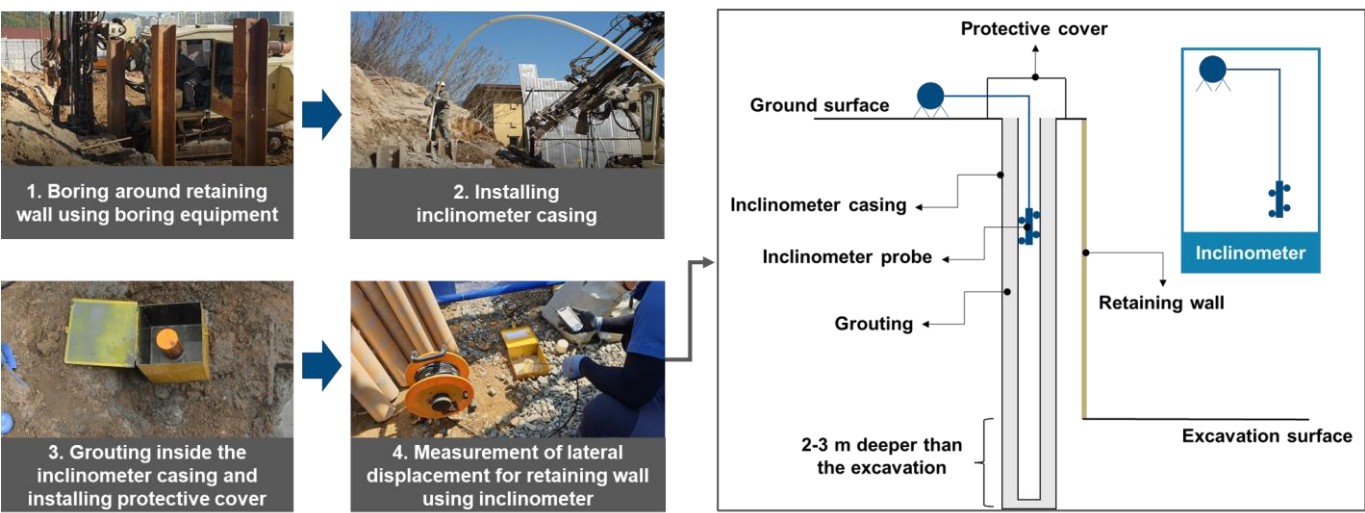

**Figure 1.** Inclinometer casing installation and measurement of the lateral displacement of the retaining wall using an inclinometer.

*2.2. Previous Studies and Technologies Related to Retaining Wall Displacement Measurement*

Previous studies and technologies addressing inclinometer problems were reviewed, as shown in Tables 1 and 2. The existing techniques are divided into contact and non-contact measurement methods. In contact measurement methods [6–11], the lateral displacement data of the retaining wall are collected using different wireless communication technologies (WiFi, LoRa, 5G, and RFID) and conventional inclinometers, while the stability of the wall is evaluated using a separate analysis module. From these methods, measurement experts can determine the lateral displacement of the retaining wall without visiting the construction site, while the construction site manager can evaluate the stability of the retaining wall in real time. Owing to these benefits, different contact measurement instruments have been commercialized and applied in the field [8–11]. However, using inclinometers in these methods has the following limitations: (1) they are difficult to install and (2) the measurement of the displacement of the retaining wall is limited to 2D local lateral displacement, which does not apply to 3D deformation.

**Table 1.** Previous studies related to retaining wall displacement measurement using contact measurement methods.

| Author/Company | Category | Object | Main Measurement Technologies | Stage |
|---|---|---|---|---|
| Ha et al., 2018 [6] | Article | Excavation retaining wall | MEMS-based inclinometer, and CDMA module | Field test |
| Chen et al., 2020 [7] | Article | Excavation retaining wall | Inclinometer, data integrator, and gateway | Field test |
| RST [8] | Product | Excavation retaining wall | MEMS inclinometer, data loggers, and gateway | Commercialization |
| Trimble [9] | Product | Retaining wall | Tiltmeter, laser-tilt sensors, data loggers, and gateway | Commercialization |
| Atmax [10] | Product | Retaining wall | Inclinometer and data loggers | Commercialization |
| Specto Technology [11] | Product | Retaining wall | Tiltmeter and data loggers | Commercialization |

**Table 2.** Previous studies related to retaining wall displacement measurement using non-contact measurement methods.

| Author | Category | Measurement Technology | Object | Stage |
|---|---|---|---|---|
| Su et al., 2006 [4] | | TLS | Retaining wall for excavation | Field test |
| Hashash et al., 2011 [2] | | TLS | Retaining wall for excavation | Field test |
| Zhao et al., 2021 [1] | | TLS | Retaining wall for excavation | Field test |
| Oskouie et al., 2016 [12] | | TLS | Retaining wall for highway | Laboratory experiments (simulation) |
| Lin et al., 2019 [13] | | TLS | Retaining wall for highway | Field test |
| Seo, 2021 [14] | | TLS | Retaining wall for tunnel | Field test |
| Aldosari et al., 2020 [16] | Article | MLS | Retaining wall for highway | Field test |
| Kalenjuk et al., 2021 [5] | | MLS | Retaining wall for highway | Field test |
| Oat et al., 2017 [3] | | Vision (photogrammetry) | Retaining wall | Laboratory experiments |
| Ko et al., 2021 [15] | | Vision (D.I.C.) | Retaining wall for excavation | Laboratory experiments |
| Ha et al., 2022 [17] | | Vision (KAZE) | Retaining wall | Laboratory experiments |

In non-contact measurement methods, the lateral displacement of the retaining wall is determined by installing the latest measurement technologies, such as TLS, mobile laser scanning (MLS), and cameras, around the retaining wall in order to solve problems related to contact measurement methods. These methods can determine the 3D displacement of the retaining wall by installing measuring instruments at fixed measurement points (TLS [1,2,4,12–14] and on cameras [3,15,17]) or moving objects such as vehicles, (MLS [5,16]) with no additional heavy equipment. The MLS-based method is suitable for horizontally long retaining walls, such as those on highways. However, it was difficult to apply the MLS-based methodology to retaining walls constructed for underground excavation, which are the focus of this study, since they are vertically installed depending upon the excavation depth. The applicability of TLS-based displacement measurement was verified by several researchers using field tests on retaining walls, which are related to excavation sites [1,2,4], highways [12,13], and tunnels [14]. In the TLS-based displacement measurement method, the displacement of the retaining wall is determined mainly by collecting point cloud data according to the measurement cycle and analyzing the differences in the distance between the point cloud data. This method is effective for measuring the displacement of retaining walls, because the error is in millimeters. However, the TLS equipment is expensive, and the measuring personnel must install and dismantle the equipment constantly because it is not suitable for continuous outdoor use. In vision-based displacement measurement, the displacement of the retaining wall is determined by examining the changes in the images according to the measurement cycle or estimating point cloud data using different vision methods, such as digital image correlation [15], photogrammetry [3], and KAZE [17]. This equipment can be permanently installed outdoors to measure the displacement of the retaining wall cost-effectively compared to the TLS equipment. However, a target plate is required to increase the displacement analysis accuracy and reduce the error to millimeters [3]. The error significantly increases when the target plate is damaged or missing. Further, the results of related studies are based on laboratory experiments, making it is difficult to apply the vision-based displacement measurement methodology to construction sites that are vulnerable to rain and snow, owing to the nature of the cameras. Hence,

the retaining wall displacement measurement system developed in this study uses a non-contact measurement method that address the limitations of conventional inclinometers. Regarding the measurement technology, a 2D LiDAR sensor that is less expensive than TLS and can continuously be used outdoors at construction sites was applied.

## 3. Development of Retaining Wall Displacement Measurement System

### 3.1. Retaining Wall Displacement Measurement Methods

For the measurement technology of the retaining wall displacement measurement hardware, a 2D LiDAR sensor, which is less costly compared to TLS and has a displacement measurement performance error on the order of millimeters [28], was selected. Figure 2 shows the measurement process of the 2D LiDAR sensor-based retaining wall displacement measurement system. First, the displacement measurement hardware was mounted under the corner strut, which was installed at the corner of the retaining wall. Then, the point cloud data of the retaining wall were collected as the construction site manager performed a 360° rotation of the 2D LiDAR sensor using a laptop-type of displacement analysis device. The collected data was stored in the displacement analysis device using wireless communication technology, while the construction site manager determined the displacement of the retaining wall by analyzing the differences in distance between the point cloud data (initial point cloud data and n-th point cloud data).

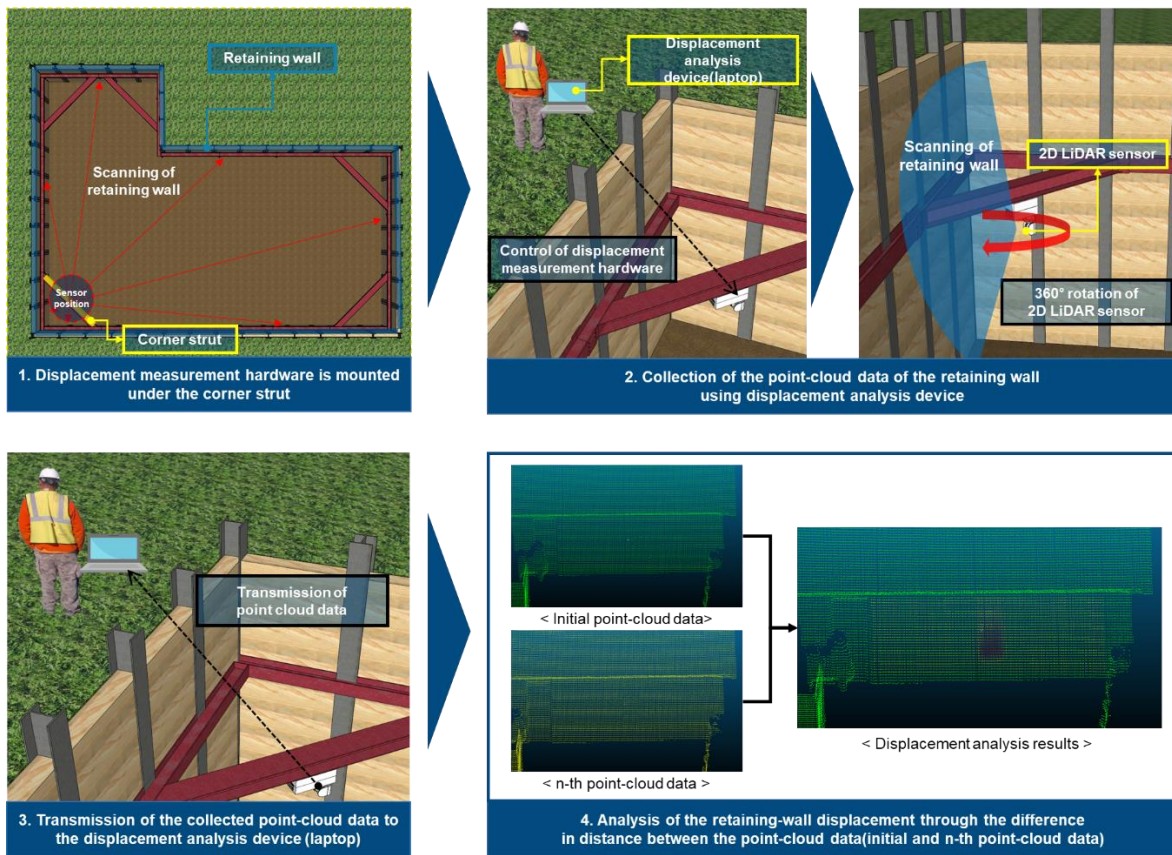

**Figure 2.** Measurement method of the 2D LiDAR sensor-based retaining wall displacement measurement system.

### 3.2. System Configuration and Design

The retaining wall displacement measurement system comprises: (1) retaining wall displacement measurement hardware mounted under the corner strut, (2) software for controlling the hardware, and (3) a displacement analysis device that stores and analyzes the data collected by the hardware. A 2D LiDAR sensor was used for displacement

analysis [28,29], and SICK's LMS511-10100 PRO product, which is more precise than other 2D LiDAR sensors, were used. Since it is a 2D line scanner, it performed only line scanning, as shown in Figure 3a and a driving device for the 360° rotation of the 2D LiDAR sensor at a constant speed (Figure 3b) was used to collect the 3D point cloud data about the retaining wall. Consequently, a servomotor (SANKYO of Japan: MM101A2LA18) and servo driver for controlling the servomotor were applied to the driving device of the hardware. Additionally, a slip ring device (COVIS of Korea: CSH025-15-1006-01E), which prevents cable twisting and allows the 2D LiDAR sensor to rotate infinitely, was applied to the hardware.

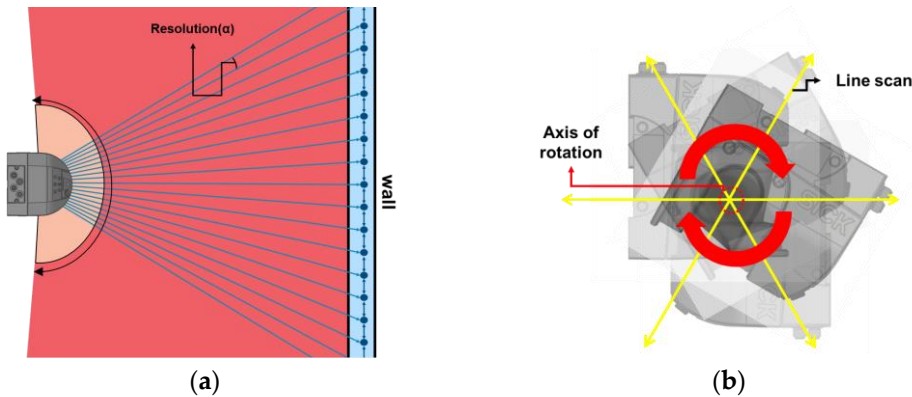

(**a**)          (**b**)

**Figure 3.** 3D point cloud data collection using the 2D LiDAR sensor. (**a**) Scan range of the 2D LiDAR sensor; (**b**) 360° rotation of the 2D LiDAR sensor.

Figure 4 shows the configuration of the 2D LiDAR sensor-based retaining wall displacement measurement system. The displacement measurement hardware receives power from an external power supply (220 V) and the power supply to the hardware (2D LiDAR sensor, servomotor, and control computer) can be confirmed through an external light-emitting diode (LED). The control computer (LATTEPANDA in China: Latte Panda Alpha 864s) controls the servomotor (RS-485 communication) and 2D LiDAR sensor (TCP/IP communication) using the software for hardware control, which stores the data collected using the 2D LiDAR sensor. The displacement analysis device (used by the construction site manager) controls the hardware by accessing the control computer remotely using WiFi and receives the point cloud data stored in the control computer.

Based on Figures 3 and 4, the displacement measurement hardware was designed, and it is depicted in Figure 5. The hardware was mounted under the corner strut using a fixing bracket, and the outer case was designed so that only the top case could be separately dismantled to allow the maintenance of the components inside the hardware. The servomotor was mounted on the right-angled reducer and set to prevent offsets from occurring when collecting the point cloud data. This is achieved by aligning the rotation axis of the reducer with the centerline of the 2D LiDAR sensor during the 360° rotation of the sensor (Figure 5d).

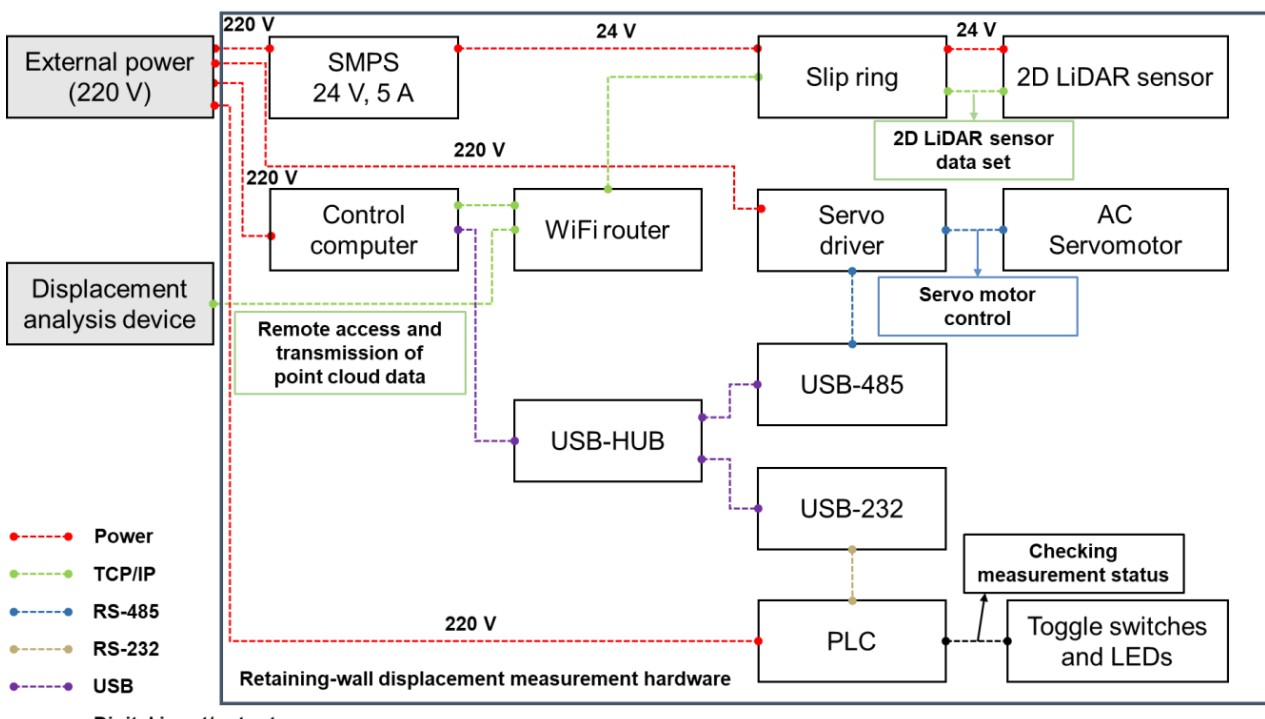

**Figure 4.** Configuration of 2D LiDAR sensor-based retaining wall displacement measurement system.

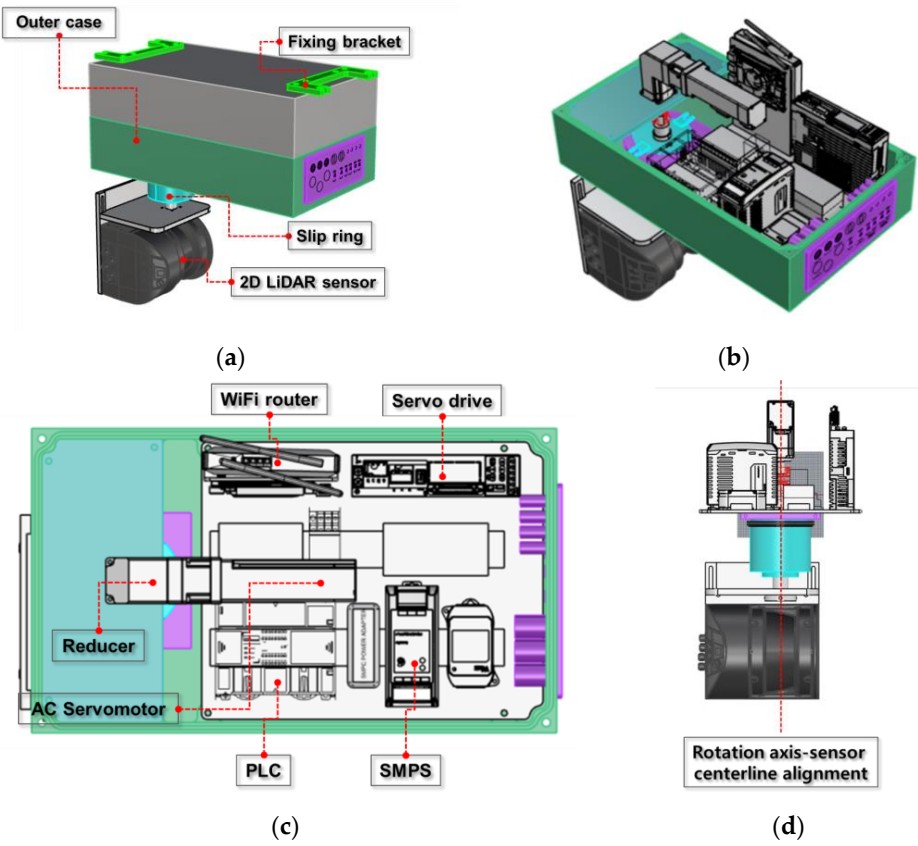

(**a**)　　　　　　　　　　　　　　　　　(**b**)

(**c**)　　　　　　　　　　　　　　　　　(**d**)

**Figure 5.** Design of the retaining wall displacement measurement hardware. (**a**) Perspective view; (**b**) perspective view inside the hardware; (**c**) layout of components inside the hardware; (**d**) alignment between the rotation axis of the reducer and centerline of the 2D LiDAR sensor.

### 3.3. Hardware and Control Software Development

From the design presented in Section 3.2, the displacement measurement hardware was developed, as shown in Figure 6. The hardware is operated by an external power supply of 220 V, while the rotation speed and number of rotations of the servomotor is manually controlled using the toggle switch inside the control panel. Additionally, the operation status of the hardware can be checked using LED.

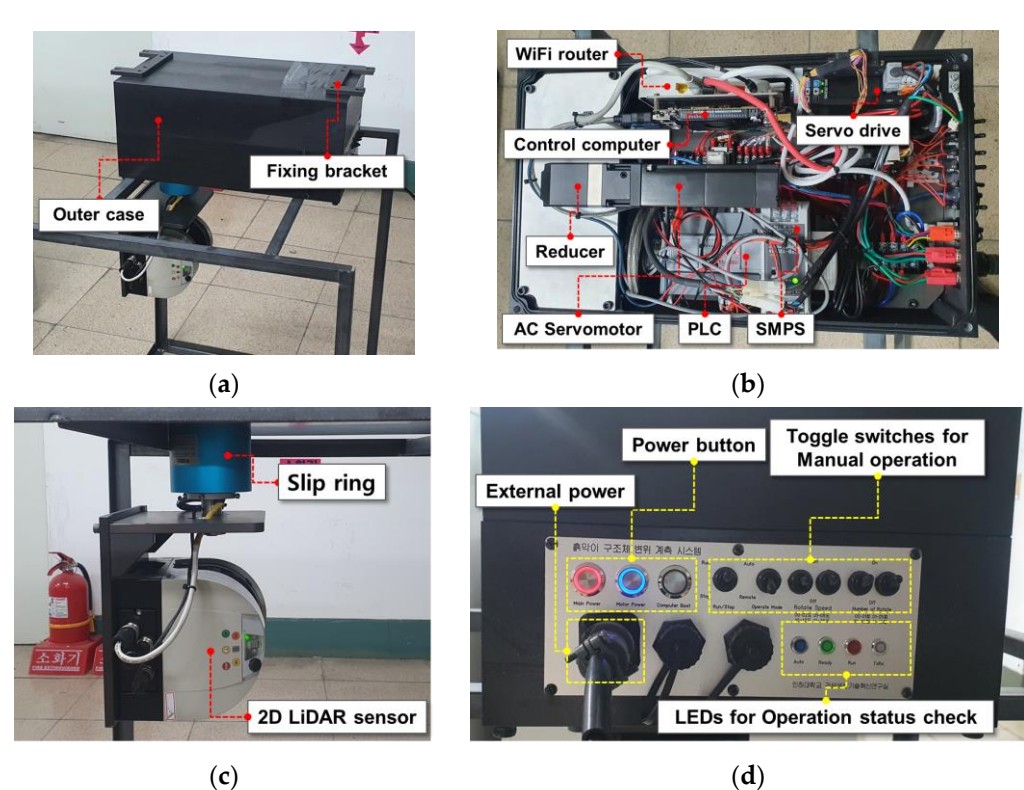

**Figure 6.** Retaining wall displacement measurement hardware. (**a**) Perspective view; (**b**) inside the hardware; (**c**) 2D LiDAR sensor and slip ring; (**d**) control panel.

When the laptop-type displacement analysis device is used, it is possible to control the hardware wirelessly using the software (as shown in Figure 7) after remotely accessing the control computer inside the hardware. The software can operate the servomotor after setting the parameters of the 2D LiDAR sensor as well as the rotation speed and number of rotations of the servomotor. Additionally, the software can check the scan pattern of the 2D LiDAR sensor in the 2D LiDAR sensor scan view during hardware operations and can monitor the current hardware status. When hardware operation is completed, the collected 3D point cloud data are automatically stored in the control computer and wirelessly transmitted to the displacement analysis device.

The 3D point cloud data collected using the displacement analysis device were saved in a comma-separated value (CSV) format, since they were stored in a data structure, as shown in Figure 8, by measuring the number of times the 2D LiDAR sensor performed line scanning. The data packet collected using the 2D LiDAR sensor comprised the distance data ($d$) and the corresponding vertical angle ($\alpha$) for each resolution set for the 2D LiDAR sensor. Since the servomotor rotates at a constant speed, the equation for converting the measurement time ($t$) and data packet of the 2D LiDAR sensor obtained using the Cartesian coordinate system ($x, y, z$) is expressed by Equation (1).

$$x = d \, sin \, \alpha \, cos \, \beta \, , \; y = d \, sin \, \alpha \, sin \, \beta \, , \; z = d \, cos \, \alpha$$
$$d = distance \, data, \alpha = vertical \, angle$$
$$\beta = horizontal \, angle = Rotation \, speed(degree/s) \times t$$

(1)

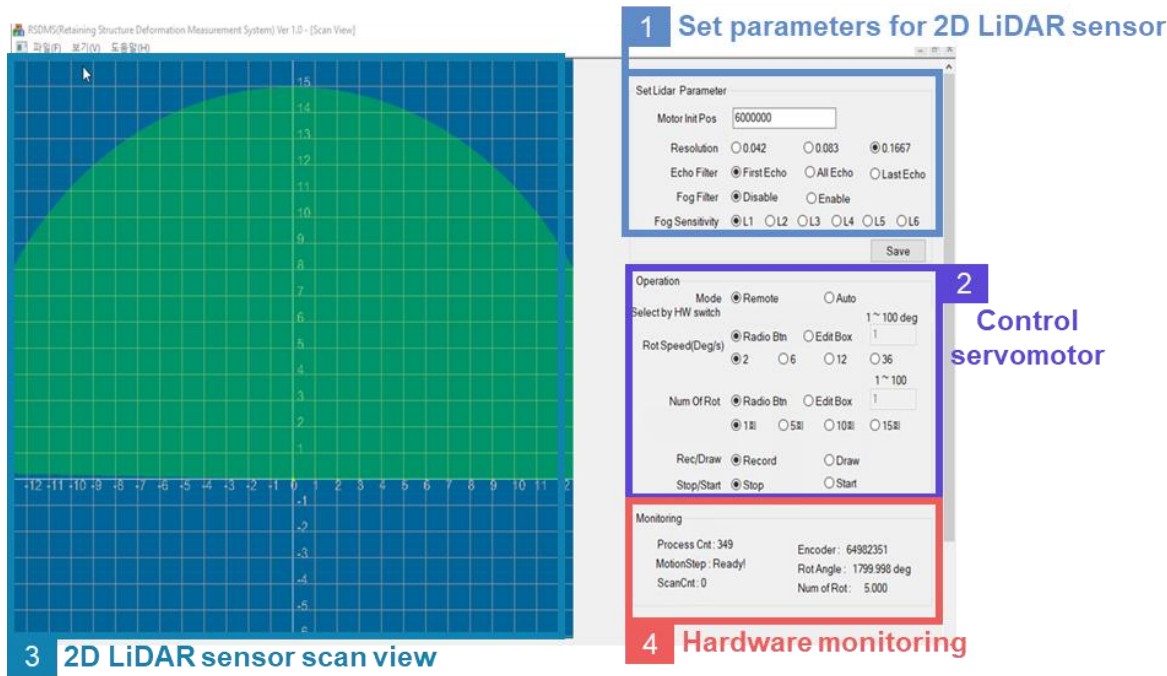

**Figure 7.** Software for hardware control.

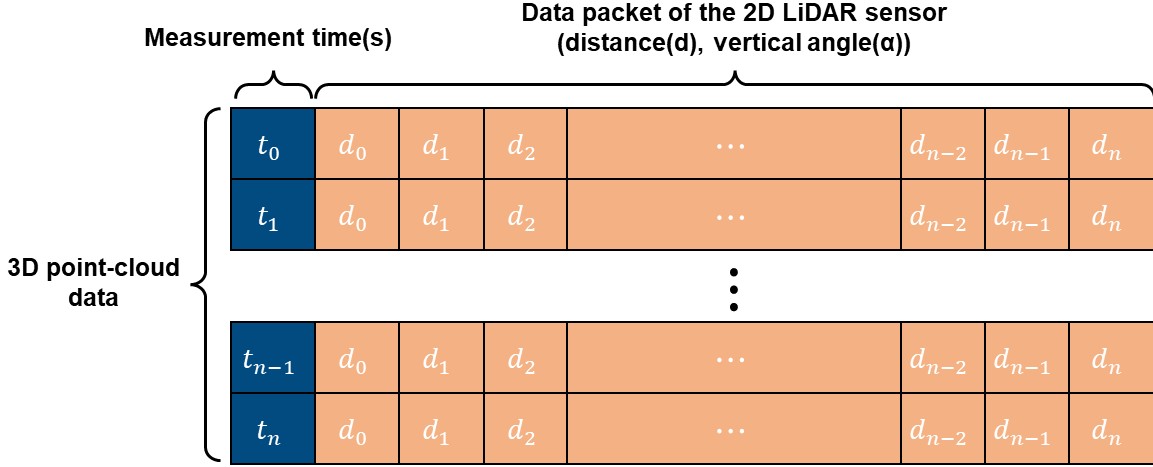

**Figure 8.** Structure of the 3D point cloud data collected by the hardware.

## 4. Performance and Economic Feasibility of a Retaining Wall Displacement Measurement System

### 4.1. System Performance

A testbed for analyzing the displacement measurement performance of the developed system was constructed, as shown in Figure 9. A laboratory experiment was performed in the following sequence: (1) collection of reference point cloud data, (2) collection of deformation point cloud data after attaching a deformation plate to the wall with the aim of causing an artificial deformation, and (3) evaluation of the distance between the reference and deformation point cloud data using a distance analysis algorithm. In the experiment, the distance between the hardware and analysis target was 20 m. The deformation of the retaining wall with a width and height of 600 and 1500 mm, respectively, was simulated by connecting five wooden plates with thicknesses of 7.15, 13.24, 20.13, 24.48, and 30.25 mm, as shown in Figure 9b.

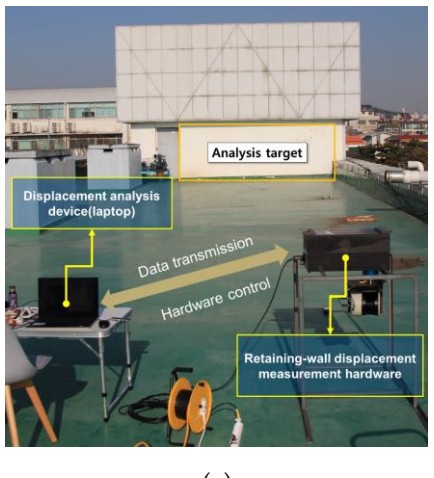
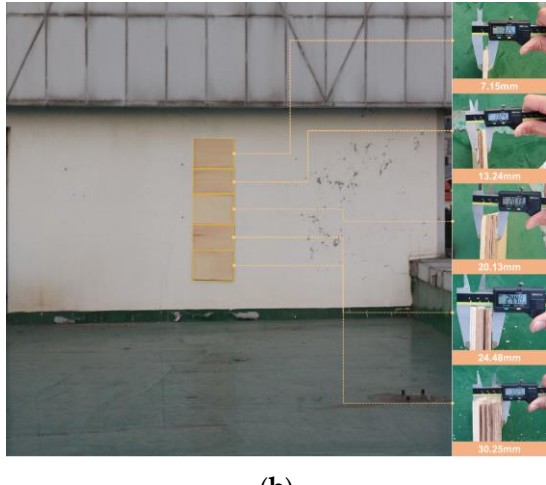

(**a**)　　　　　　　　　　　　　　　　　(**b**)

**Figure 9.** Testbed for evaluating the performance of the retaining wall displacement measurement system. (**a**) Scene of the performance test; (**b**) simulation of retaining wall deformation.

The main parameters of the displacement measurement hardware were set as shown in Table 3. The Multiple Model-to-Model Cloud Comparison (M3C2) algorithm, which is sensitive to sensor noise and more accurate than other algorithms (C2C and C2M), was used to evaluate the distances between the point cloud data. The M3C2 algorithm developed by Lague et al. [30] selected core points from the reference point cloud as distance analysis targets and set the direction of deformation by generating the normal vector of the plane, which was fitted using the neighboring points, and included the diameter $D_{M3C2}$ for each core point. A cylinder with diameter $d_{M3C2}$ and height $H_{M3C2}$ was then created according to the normal vector direction of individual core points. The reference and deformation point cloud data inside the cylinder were projected onto the normal vector, and the average distances between the projected data were calculated. Table 3 presents the parameters of the M3C2 algorithm used in this study. The displacement measurement performance of the developed system was evaluated using the root-mean-square error (RMSE) as expressed by Equation (2).

$$RMSE(Root\ Mean\ Square\ Error) = \sqrt{\frac{\sum_{i=1}^{n}(PV_i - GT_i)^2}{n}} \tag{2}$$

GT = Ground Truth, PV = Predicted Value

**Table 3.** Main parameters of the retaining wall displacement measurement hardware and M3C2 algorithm.

| Parameter | | Value |
|---|---|---|
| 2D LiDAR sensor | Resolution | 0.1667° |
| | Frequency | 25 Hz |
| Servomotor | Rotation speed | 2°/s |
| | Number of rotations | 1 |
| M3C2 algorithm | Core points | All point cloud data |
| | $D_{M3C2}$ (normal scale) | 300 mm |
| | $d_{M3C2}$ (projection scale) | 300 mm |
| | $H_{M3C2}$ (height) | 3000 mm |

Figure 10 shows the reference and deformation point cloud data collected using testbed. Figure 11 shows the ground truth of lateral displacement and the displacement analysis results based on the M3C2 algorithm. The ground truth in the wooden plate position in the analysis target is the thickness of the wooden plate, and in the other areas, the ground

truth is set to 0. The *x*-axis shows the wall width (width of the analysis target), while the *y*-axis shows the wall height (height of the analysis target). The color bar made of red and blue shows the lateral displacement. In the displacement analysis results, the lateral displacement gradually decreased as the wall height increased, which is like the ground truth, while the displacement measurement position was like the ground truth. Additionally, the deformation of approximately 7.15 mm was considerably recognizable. Since the RMSE between the ground truth and the predicted lateral displacement was 2.82 mm (number of analysis points: 5114), it was concluded that the developed displacement measurement system had an error on the order of millimeters. Also, because the system is very accurate in detecting lateral displacement, which exceeds the Korean threshold of 9 mm (for an excavation depth of 3 m) [24], it is highly applicable in the field.

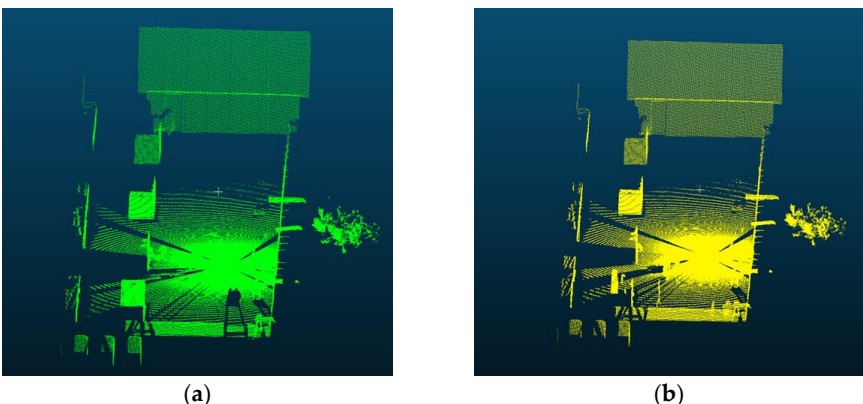

(**a**)                                                                                                       (**b**)

**Figure 10.** Reference and deformation point cloud data collected. (**a**) Reference point cloud data; (**b**) deformation point cloud data.

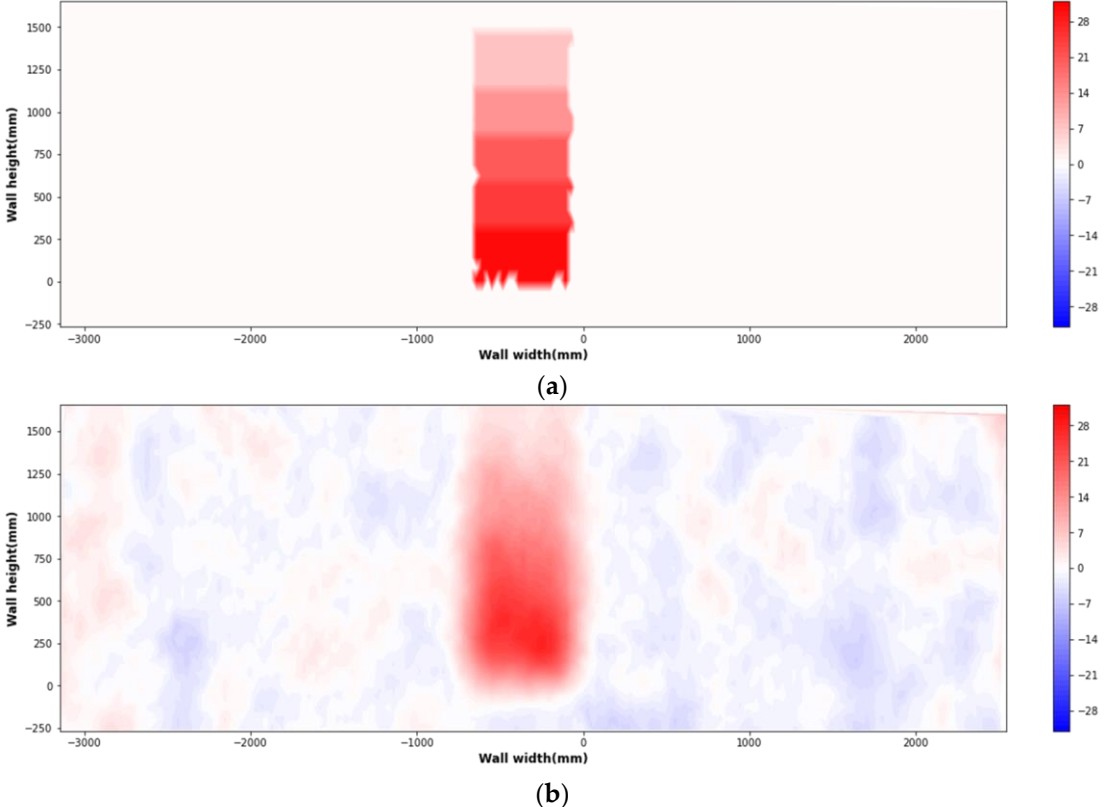

**Figure 11.** Distance analysis results obtained using the M3C2 algorithm. (**a**) Ground truth of lateral displacement; (**b**) predicted values of lateral displacement (RMSE: 2.82 mm).

### 4.2. Economic Feasibility of System

The retaining wall displacement measurement system with an error on the order of millimeters was developed in order to address conventional inclinometer limitations, and its performance evaluated. However, if the economic feasibility of the system is not ensured, continuous research and development will be limited, thereby hampering the commercialization of the system and its introduction to construction sites upon its developmental completion. Therefore, it is necessary to analyze the economic feasibility of the system using different techniques. The process used for analyzing the economic feasibility of the developed system is shown in Figure 12. It comprises the following steps: (1) defining the lifecycle cost (LCC) analysis scope, (2) setting the LCC analysis variables and assumptions, (3) economic analysis, and (4) sensitivity analysis.

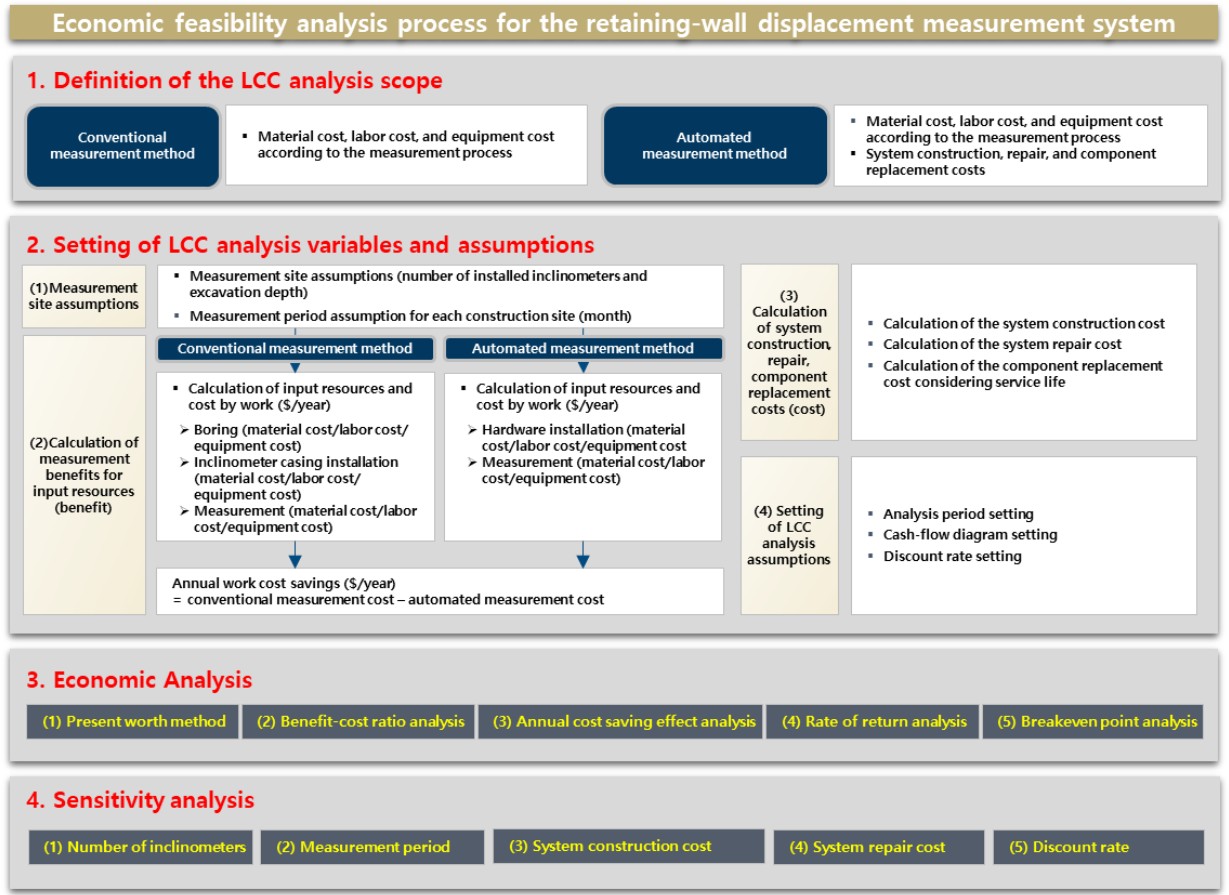

**Figure 12.** Economic feasibility analysis process.

#### 4.2.1. Definition of Lifecycle Cost Analysis Scope

The LCC analysis scope was defined by analyzing the conventional measurement process (Figure 1) and automated measurement process (Figure 2), as shown in Figure 13. The conventional measurement method has labor, material, and equipment costs owing to the manpower, equipment, and material inputs necessary for boring, inclinometer casing installation, and measurements. Regarding the automated measurement method, the labor, material, and equipment costs are associated with the installation of the displacement measurement hardware and measurement. Since hardware installation is easier than that in conventional methods, and measurements are performed by a single worker, economic benefits can be seen when the input of personnel and equipment are reduced. The automated measurement method has the following additional costs: (1) system construction, (2) system repair, and (3) replacement costs for components that have expired.

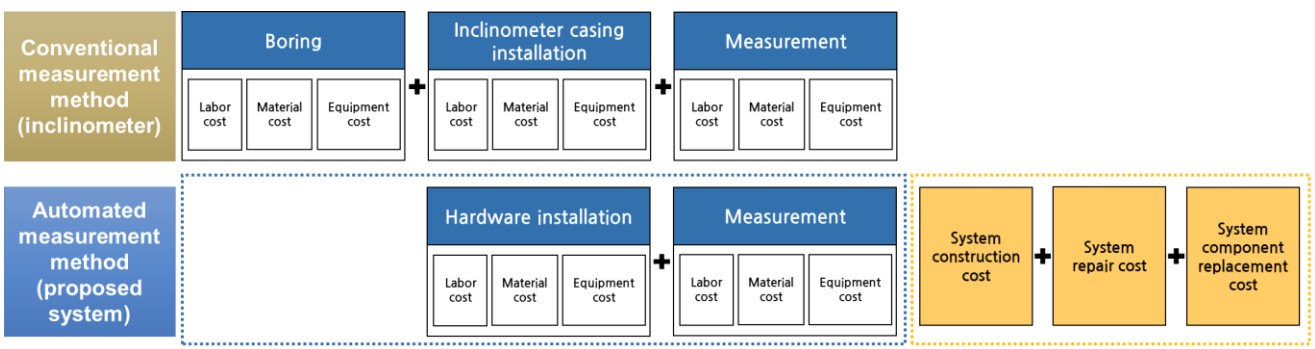

**Figure 13.** Life cycle cost (LCC) analysis scope.

4.2.2. Setting of LCC Analysis Variables and Assumptions

Since different construction sites have different measurement plans, the measurement site was assumed, as shown in Figure 14 and Table 4, in order to analyze the LCC for the conventional and automated measurement methods under similar conditions. In the conventional method, the measurement cost increased with the excavation depth. However, the excavation depth was set at 3 m for a conservative economic analysis. Further, it was assumed that displacement measurements on both sides of the retaining wall were performed using two inclinometers. An interview with a company that specialized in measurement revealed that the measurement period for small- and medium-sized construction sites with excavation depths of <10 m was six months. Measurements were conducted twice weekly over a six-month period based on the technical guidelines on measurement and management for excavation construction in Korea, for a total of 48 measurements. To evaluate the variability of the LCC analysis results under the aforementioned conditions, a sensitivity analysis was conducted on the assumptions for the number of inclinometers and measurement period, which are expected to have a considerable variability among construction sites.

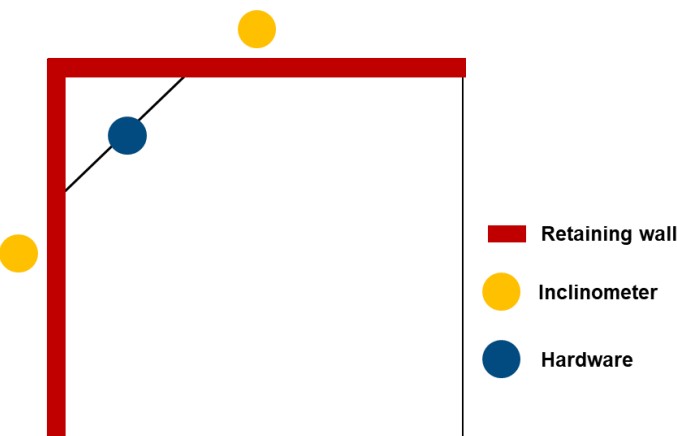

**Figure 14.** Measurement-site assumptions for LCC analyses.

The work cost (labor, material, and equipment) of the conventional measurement method was calculated using the standard quantity per unit of Korea's ground survey published by the Korea Engineering and Consulting Association [31]. Since the hardware must be mounted under the corner strut for the automated measurement method, the input of one corner strut installation person (ordinary worker, steelworker, or welder) and equipment (tower crane) was applied [32]. The quantity was then calculated with the input of an intermediate technician, which is similar to the calculation for the inclinometer casing installation work. Since the retaining wall displacement measurement work can be performed more conveniently compared to the conventional method, the quantity was

calculated with the input of a beginner technician in a similar manner as the calculation for one inclinometer measurement. Regarding the unit price for the above quantity, the labor cost was derived from the wage survey report of Korean engineering companies [33], the report on the wage status of Korean construction industries [34], the material cost based on the price information magazine of Korea [35] and estimates from related companies, as well as the equipment cost based on the cost calculation table for construction machinery in Korea [36]. The work costs for the conventional and automated measurement methods (Tables 5–7) were calculated as $9870.93 and $1466.85, respectively, showing that the cost benefit of the automated measurement method compared to the conventional measurement method was $8404.08 per year.

**Table 4.** Assumed measurement site parameters.

| Parameters | |
| --- | --- |
| Excavation depth | 3 m |
| Boring depth (Excavation depth + 2 m) | 5 m |
| Inclinometer | 2 units |
| Hardware | 1 unit |
| Measurement period | 6 months |
| Number of measurements | 48 |

**Table 5.** Boring cost for the conventional method.

| Input Cost | | | Conventional Method | |
| --- | --- | --- | --- | --- |
| | Item | Unit Cost | Quantity | Cost |
| Labor cost | Intermediate technician | $160.95/person | 0.12 person | $22.53 |
| | Special worker | $132.33/person | 0.17 person | $25.14 |
| | Ordinary worker | $105.46/person | 0.23 person | $28.47 |
| | Borer | $141.36/person | 0.23 person | $38.17 |
| | Construction machine operator | $168.06/person | 0.11 person | $18.49 |
| | Labor cost per unit boring depth ($/m) | | $122.02/m | |
| Material cost | Single-core barrel (HX × 1.0 m) | $138.69/box | 0.01 box | $1.39 |
| | Metal crown bit (HX) | $37.96/box | 0.025 box | $0.95 |
| | Drive pipe head (HX) | $43.80/box | 0.01 box | $0.44 |
| | Drive pipe shoe (HX) | $36.50/box | 0.01 box | $0.36 |
| | Drive pipe (HX × 1.0 m) | $90.51/box | 0.01 box | $0.91 |
| | Material cost per boring depth ($/m) | | $4.04/m | |
| Equipment cost | Boring equipment (54HP) | $28.55/h | 0.88 h | $25.13 |
| | Equipment cost per unit boring depth ($/m) | | $25.13/m | |
| Work cost per unit boring depth ($/m) = labor cost + material cost + equipment cost | | | $151.19/m | |
| Boring cost ($) = work cost per unit boring depth × boring depth × number of inclinometers | | | $1511.92 | |

The system construction cost for the automated measurement method was calculated as the cost invested in constructing the system (Table 8), while the system repair cost was calculated to be 5% of the system's construction cost every year. The service life of each component was determined based on the service life table of the Public Procurement Service of Korea [37] and interviews with system manufacturers, while the component replacement cost was set by considering an entire replacement depending on the service life. To evaluate the variability of the LCC analysis results based on the changes in the system input cost, a sensitivity analysis was conducted regarding the changes in system construction and repair costs.

**Table 6.** Costs for the inclinometer casing (conventional method) and hardware installations (automated method).

| Input Cost | | | Conventional Method | | Automated Method | |
|---|---|---|---|---|---|---|
| | Item | Unit Cost | Quantity | Cost | Quantity | Cost |
| Labor cost | Intermediate technician | $169.18/person | 0.40 person | $67.67 | 0.40 person | $67.67 |
| | Special worker | $140.42/person | 0.80 person | $112.34 | - | $0 |
| | Ordinary worker | $112.17/person | 1.60 person | $179.47 | 0.13 person | $14.58 |
| | Intermediate skilled technician | $142.07/person | 0.80 person | $113.66 | - | $0 |
| | Steelworker | $158.18/person | - | $0 | 0.34 person | $53.78 |
| | Welder | $174.26/person | - | $0 | 0.17 person | $29.62 |
| | Construction machine operator | $168.06/person | - | $0 | 0.04 person | $6.72 |
| | Labor cost per installed unit ($/number of units) | | $473.13/number of units | | $172.38/number of units | |
| Material cost | Inclinometer casing (Boring depth + 1 m) | $8.76/m | 6 m | $52.55 | - | $0 |
| | Protective box | $72.99/box | 1 box | $72.99 | - | $0 |
| | Miscellaneous materials (% of the sum of inclinometer casing and protective box) | $125.55 | 5% | $6.28 | - | $0 |
| | Material cost per installed unit ($/number of units) | | $131.82/number of units | | $0/number of units | |
| Equipment cost | Crane | $51.72/h | - | $0 | 0.29 h | $15.00 |
| | Equipment cost per installed unit ($/number of units) | | $0/number of units | | $15.00/number of units | |
| Work cost per installed unit ($/number of measuring instruments) = labor cost + material cost + equipment cost | | | $604.96/number of units | | $187.38/number of units | |
| Installation cost ($) = work cost per installed unit × number of measuring instruments | | | $1209.92 | | $187.38 | |

**Table 7.** Measurement costs for the conventional and automated methods.

| Input Cost | | | Conventional Method | | Automated Method | |
|---|---|---|---|---|---|---|
| | Item | Unit Cost | Quantity | Cost | Quantity | Cost |
| Labor cost | Intermediate technician | $169.18/person | 0.10 person | $16.92 | - | $0 |
| | Special worker | $140.42/person | 0.20 person | $28.08 | - | $0 |
| | Beginner technician | $133.28/person | 0.20 person | $26.66 | 0.20 person | $26.66 |
| | Labor cost per measurement | | $71.66 | | $26.66 | |
| Material cost | Cost of using the inclinometer | $14.06 | 0.20 | $2.81 | - | $0 |
| | Material cost per measurement | | $2.81 | | $0 | |
| Work cost per measurement ($/number of measurements) = labor cost + material cost | | | $74.47/number of measurements | | $26.66/number of measurements | |
| Inclinometer installation cost ($) = work cost per measurement × number of measurements × number of measuring instruments | | | $7149.09 | | $1279.47 | |

The LCC analysis period was set at 10 years, considering the system component with the longest service life. Accordingly, the cash flow diagram was derived, as shown in Figure 15. The inflation rate in Korea over the past 10 years [38] and the bank deposit interest rate [39] were used as discount rates for the LCC analysis. The average inflation rate over the past 10 years was used for the LCC analysis, and a sensitivity analysis was conducted regarding the discount rate for the inflation and bank deposit interest rates range (0.90%–3.43%) over the past 10 years (Figure 16).

**Table 8.** Construction cost for retaining wall displacement measurement systems and component costs.

| | Product | Service Life (Years) | Cost ($) |
|---|---|---|---|
| Component | 2D LiDAR sensor (LMS511-10100 PRO) | 10 | $5474.45 |
| | Bracket | 10 | $218.98 |
| | Servomotor | 10 | $306.57 |
| | Reducer | 10 | $291.97 |
| | Slip ring | 5 | $255.47 |
| | Servomotor driver | 5 | $109.49 |
| | PLC | 10 | $218.98 |
| | Control computer | 6 | $656.93 |
| | Control and drive unit case | 10 | $729.93 |
| | Power | 10 | $145.99 |
| | Bolts and other consumables | 1 | $291.97 |
| | Displacement analysis device | 6 | $1094.89 |
| Assembly cost | | - | $1094.89 |
| Test cost | | - | $1094.89 |
| Construction cost of the retaining wall displacement measurement system | | | $11,985.40 |

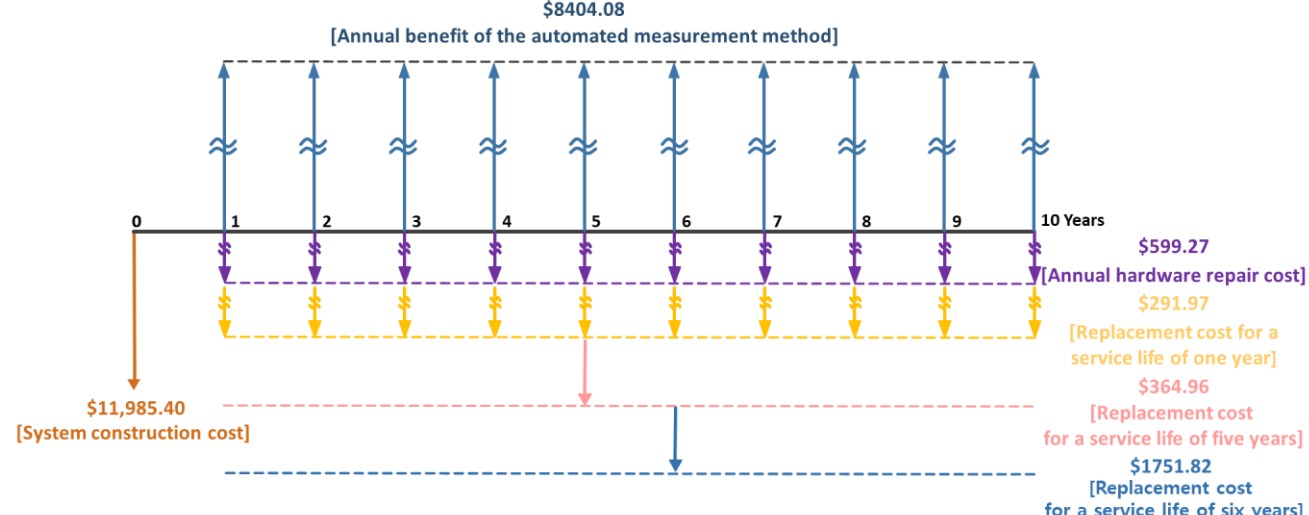

**Figure 15.** Cash flow diagram for the LCC analysis.

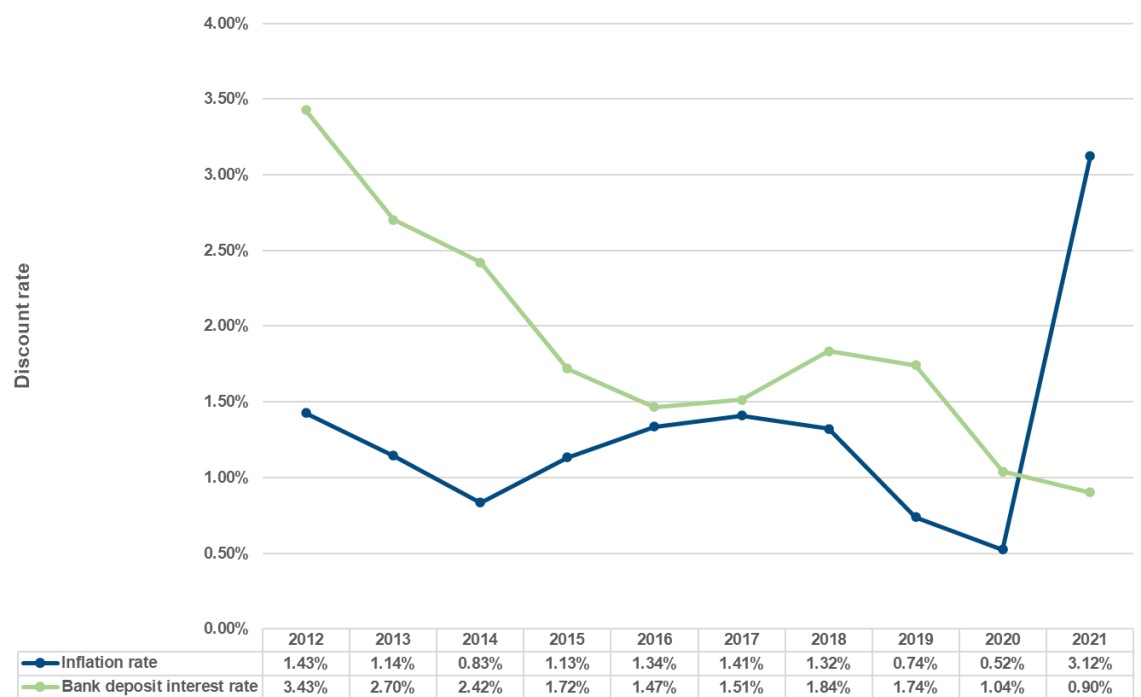

| | 2012 | 2013 | 2014 | 2015 | 2016 | 2017 | 2018 | 2019 | 2020 | 2021 |
|---|---|---|---|---|---|---|---|---|---|---|
| Inflation rate | 1.43% | 1.14% | 0.83% | 1.13% | 1.34% | 1.41% | 1.32% | 0.74% | 0.52% | 3.12% |
| Bank deposit interest rate | 3.43% | 2.70% | 2.42% | 1.72% | 1.47% | 1.51% | 1.84% | 1.74% | 1.04% | 0.90% |

**Figure 16.** Inflation and bank deposit interest rates in Korea over the past 10 years (2012–2021).

### 4.2.3. Economic Analysis

According to the cash flow diagram and discount rate, an economic analysis was performed, which included the following: (1) present worth, (2) benefit/cost (B/C) ratio, (3) annual measurement cost-saving effect, (4) rate of return (ROR), and (5) breakeven point analyses (Table 9). The calculation formulas used for the economic analysis are presented in Appendix A.

**Table 9.** Economic analysis definitions.

| Economic Analysis | Definition |
|---|---|
| Present worth analysis | A method of comparing benefits and costs by converting all the benefits and costs in the cash flow diagram into the present worth. |
| B/C ratio analysis | A method of evaluating the degree to which the benefit exceeds the cost by analyzing the ratio of the benefit to the cost converted into the present worth. |
| Annual measurement cost-saving effect analysis | A method of identifying the annual measurement cost-saving effect of introducing a system by comparing the annual benefit and cost under the assumption that the benefit and cost are identical in different years. |
| ROR analysis | A method of deriving the rate of return that differentiates between the benefit and cost converted into the present worth zero. |
| Breakeven point analysis | A method of deriving the breakeven point between the benefit and cost converted into the present worth. |

According to the economic analysis results (Table 10), the system has a present worth of $56,075.91, a B/C ratio of 3.52, an annual measurement cost-saving effect of 71.59%, an ROR of 61.48%, and a breakeven point of 2.71 years. The present worth was positive, and the B/C ratio was > 1, showing that the automated measurement method is more economically feasible than the conventional measurement method. When the automated measurement method is adopted, the annual measurement cost is reduced by 71.59% compared to the conventional method, and a net profit is generated after 2.71 years by recovering all investments in system construction.

**Table 10.** Economic analysis results.

| Present Worth | B/C Ratio | Annual Measurement Cost-Saving Effect | ROR | Breakeven Point |
| --- | --- | --- | --- | --- |
| $56,075.91 | 3.52 | 71.59% | 61.48% | 2.71 years |

### 4.2.4. Sensitivity Analysis

To improve the reliability of the LCC analysis, it was necessary to evaluate the variability of the economic analysis results by changing the LCC analysis variables and assumptions. Consequently, a sensitivity analysis (Appendix B) was conducted by changing five variables (the number of inclinometers, measurement period, system construction cost, system repair cost, and discount rate [Table 11]).

**Table 11.** Ranges of the variables subjected to sensitivity analysis.

| Variable | Range |
| --- | --- |
| Number of inclinometers | 1–4 |
| Measurement period | 2–12 months |
| System construction cost | 60–140% of the existing construction cost |
| System repair cost | 5–20% of the system construction cost every year |
| Discount rate | 0.52–3.43% |

The sensitivity analysis results (Figure 17) showed that the developed system was economically feasible within the analysis range of all five variables. The yellow bar in Figure 17 shows the results of the economic analysis in Section 4.2.3. Among the five variables, the economic feasibility of the system was the lowest (present worth = $10,074.09, B/C ratio = 1.45, annual measurement cost-saving effect = 31.16%, ROR = 15.02%, and breakeven point = 6.74 years) when the number of inclinometers was 1. When a single inclinometer was installed, measurements were performed only for one side of the retaining wall, and the probability of this case was low. Nevertheless, the retaining wall displacement measurement system had economic benefits relative to the conventional measurement method.

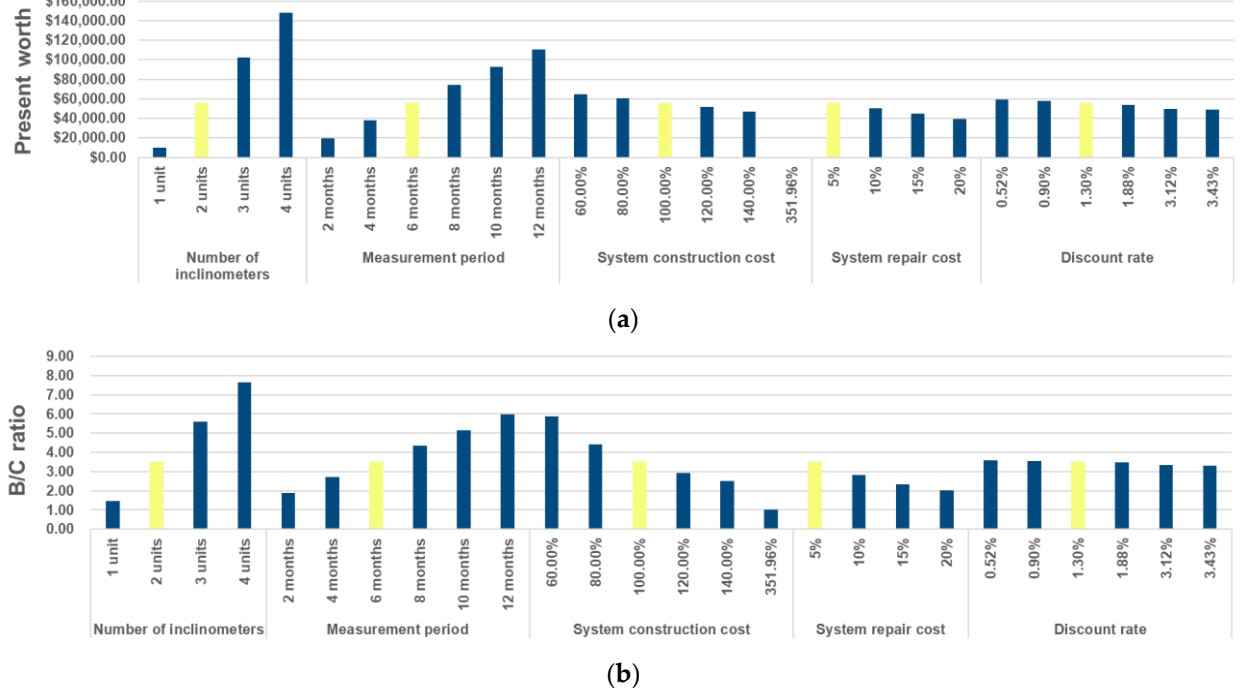

**Figure 17.** *Cont.*

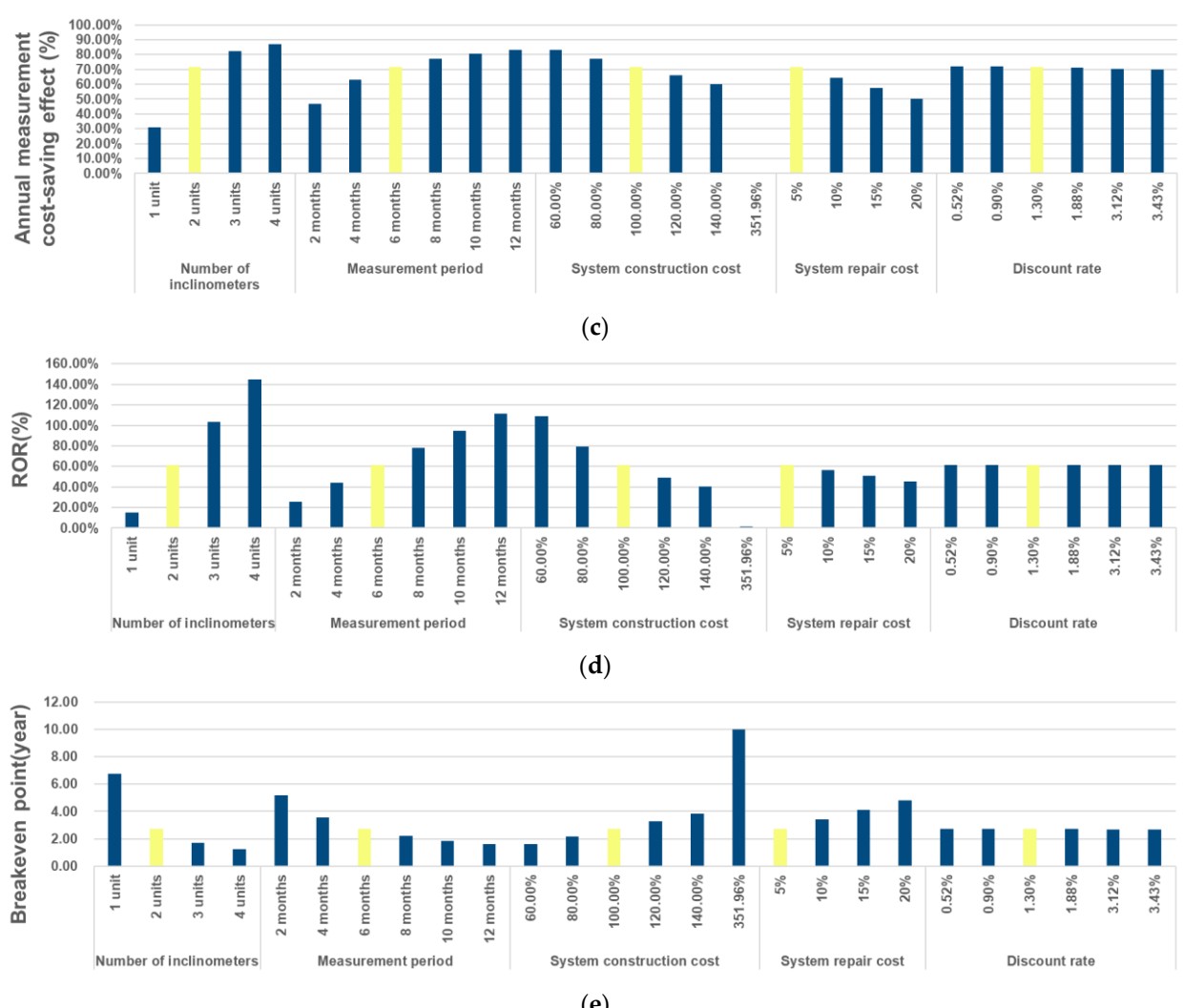

**Figure 17.** Sensitivity analysis results regarding the LCC analysis variables and assumptions. (**a**) Present worth analysis; (**b**) benefit/cost (B/C) ratio analysis; (**c**) annual measurement cost-saving effect analysis; (**d**) ROR analysis; (**e**) breakeven point analysis.

Among the variables considered, the number of inclinometers had the most considerable effect on the economic feasibility. Regarding a site where four inclinometers were installed, the ROR of the retaining wall displacement measurement system increased to 144.87%. The economic feasibility of the system increased as the measurement period increased, showing that the manpower input was larger for the conventional measurement method, since it was based on manpower. The effects of changes in the system construction and system repair costs, as well as the discount rate on the economic analysis results were relatively insignificant, showing that the economic feasibility of the system was not sensitive to these variables. Since the economic feasibility of the system aggravates when system construction costs increase by 351.96% ($42,183.82), the system must be designed and constructed to avoid exceeding this cost level, especially when the cost increases owing to system improvement.

## 5. Conclusions

Methods for measuring retaining wall displacements using inclinometers had the following problems: difficulties in installing them, in identifying the 2D local lateral displacement of the retaining wall, and in measuring using manpower. In order to address these problems, a 2D LiDAR sensor-based retaining wall displacement measurement system

was developed in this study. The performance of the developed system was evaluated, and an economic analysis was performed in order to analyze the reliability of the measurement results and the economic feasibility of the system. The following conclusions were drawn.

1.  The retaining wall displacement measurement system comprised: (1) a retaining wall displacement measurement hardware, (2) a software for controlling the hardware, and (3) a displacement analysis device. As compared to TLS, a 2D LiDAR sensor which is economical was applied to the retaining wall displacement measurement hardware. This hardware was developed to collect the point cloud data of the retaining wall by rotating the 2D LiDAR sensor at 360° at a constant speed using a servomotor. The hardware was designed so that wireless control can be performed using software for hardware control, and a displacement analysis device.

2.  A testbed was constructed to evaluate the displacement measurement performance of the developed system. A deformation of approximately 7.15 mm was considerably recognizable and the RMSE between the ground truth and lateral displacement predicted using the developed system was 2.82 mm. The system was considerably accurate with Korea's lateral displacement threshold, thus revealing a high applicability in the construction sector.

3.  The results of the LCC analysis revealed that the developed system had a present worth of $56,075.91, a B/C ratio of 3.52, an annual measurement cost-saving effect of 71.59%, an ROR of 61.48%, and a breakeven point of 2.71 years, revealing that it had high economic feasibility. The sensitivity analysis results showed that the developed system was economically feasible in the analysis ranges of five selected variables. Among the five variables, the economic feasibility of the system was lowest when the number of inclinometers was 1. The effects of the system construction and system repair costs, as well as the discount rate on the variability of the economic analysis results were insignificant. Since the economic feasibility of the system aggravated when the system construction cost increased by 351.96% ($42,183.82), the commercialization model of the system had to be developed to avoid exceeding this cost level.

This study however had a limitation: the displacement measurement performance of the system was evaluated using a discontinuous deformation simulation model. Hence, it is necessary to verify the displacement measurement performance and set the rotation speed and number of rotations by constructing a testbed that can simulate the continuous deformation of an actual retaining wall. In the future, displacement analysis software that can easily analyze and confirm the results will be developed. If the developed system is used for retaining wall displacement measurements—particularly at small- and medium-sized construction sites—it is expected that problems with the conventional manpower-based measurement methods will be resolved. Additionally, the applicability of the system is expected to considerably improve using the economic feasibility analysis results obtained in this study.

**Author Contributions:** Conceptualization, J.-S.K. and Y.S.K.; Data curation, J.-S.K. and G.-y.L.; Investigation, J.-S.K.; Methodology, J.-S.K. and G.-y.L.; Project administration, Y.S.K.; Validation, J.-S.K. and Y.S.K.; Visualization, J.-S.K.; Writing—original draft, J.-S.K.; Writing—review & editing, G.-y.L. and Y.S.K. All authors have read and agreed to the published version of the manuscript.

**Funding:** This work was supported by the National Research Foundation of Korea (NRF) grant funded by the Korean government (MSIT) (No. 2020R1A2C2008616).

**Data Availability Statement:** Data is contained within the article.

**Conflicts of Interest:** The authors declare no conflict of interest.

## Appendix A

$$\text{Present Worth (PW)} = \text{Present Worth of Benefit} - \text{Present Worth of Cost}$$
$$= \text{Annual benefit of the automated measurement method } (P/A, \text{ i, n})-$$
$$(\text{system construction cost} + \text{ annual hardware repair cost } (P/A, \text{ i, n})+$$
$$\text{present worth of the component replacement cost})$$
$$= \$8404.08(P/A, \ 1.30\%, 10)-$$
$$\{\$11,985.40 + \$599.27(P/A, 1.30\%, 10) + \$291.97(P/A, 1.30\%, 10)$$
$$+\$364.96(P/F, 1.30\%, 5) + \$1751.82(P/F, 1.30\%, 6)\}$$
$$= \$78,332.60 - (\$11,985.40 + \$5585.60 + \$2721.36 + \$342.14 + \$1621.19) = \$56,076.91$$

$$\text{Benefit/cos t ratio (B/C ratio)} = \frac{\text{Present Worth of Benefit}}{\text{Present Worth of Cost}}$$
$$= \frac{\text{Annual benefit of the automated measurement method } (P/A, \text{ i, n})}{(\text{system construction cost} + \text{ annual hardware repair cost } (P/A, \text{ i, n}) + \text{present worth of the component replacement cost})}$$
$$= \frac{\$8404.08(P/A, \ 1.30\%, 10)}{\{\$11,985.40 + \$599.27(P/A, 1.30\%, 10) + \$291.97(P/A, 1.30\%, 10) + \$364.96(P/F, 1.30\%, 5) + \$1751.82(P/F, 1.30\%, 6)\}}$$
$$= \frac{\$78,332.60}{\$22,255.69} = 3.52$$

$$\text{Annual measurement cost saving effect}$$
$$= -\frac{\text{Equivalent Uniform Annual Cost} - \text{Equivalent Uniform Annual Benefit}}{\text{Equivalent Uniform Annual Benefit}}$$
$$= -\frac{\$22,255.69(A/P, 1.96\%, 40) - \$8404.08}{\$8404.08} = -\frac{\$2387.78 - \$8404.08}{\$8404.08}$$
$$= 71.59\%$$

Rate of Return (i) : Find the interest (i) at which the present worth of benefit and present worth of cost are equal
Present Worth of Benefit(i) = Present Worth of Cost(i)
$= \$8404.08(P/A, \text{ i}, 10)$
$-\{\$11,985.40 + \$599.27(P/A, \text{i}, 10) + \$291.97(P/A, \text{i}, 10) + \$364.96(P/F, \text{i}, 5) + \$1751.82(P/F, \text{i}, 6)\} = 0$
$\text{i} \approx 61.48\%$

Breakeven point (n) : Find the year (n) at which the present worth of benefit and present worth of cost are equal
Present Worth of Benefit(n) = Present Worth of Cost(n)
$\$8404.08(P/A, \ 1.30\%, \text{n}) = \$11,985.40 + \$599.27(P/A, 1.30\%, 10)$
$+\$291.97(P/A, 1.30\%, 10) + \$364.96(P/F, 1.30\%, 5) + \$1751.82(P/F, 1.30\%, 6)$
$(P/A, \ 1.30\%, \text{n}) = \$22,255.69/\$8404.08 = 2.65$
$\text{n} \approx 2.71 \text{ years}$

## Appendix B

**Table A1.** Sensitivity analysis results according to the number of inclinometers.

| Number of Inclinometers | Current Worth Method | Benefit/Cost Ratio | Annual Work cost Reduction Rate | Rate of Return | Breakeven Point |
|---|---|---|---|---|---|
| 1 | $10,074.09 | 1.45 | 31.16% | 15.02% | 6.74 years |
| 2 (reference) | $56,075.91 | 3.52 | 71.59% | 61.48% | 2.71 years |
| 3 | $102,077.73 | 5.59 | 82.10% | 103.47% | 1.70 years |
| 4 | $148,079.54 | 7.65 | 86.93% | 144.87% | 1.24 years |

**Table A2.** Sensitivity analysis results according to the measurement period.

| Measurement Period | Current Worth Method | Benefit/Cost Ratio | Annual Work Cost Reduction Rate | Rate of Return | Breakeven Point |
|---|---|---|---|---|---|
| 2 months | $19,603.39 | 1.88 | 46.83% | 25.81% | 5.16 years |
| 4 months | $37,839.65 | 2.70 | 62.97% | 44.23% | 3.56 years |
| 6 months (reference) | $56,075.91 | 3.52 | 71.59% | 61.48% | 2.71 years |
| 8 months | $74,312.17 | 4.34 | 76.95% | 78.27% | 2.19 years |
| 10 months | $92,548.43 | 5.16 | 80.61% | 94.85% | 1.84 years |
| 12 months | $110,784.69 | 5.98 | 83.27% | 111.33% | 1.59 years |

**Table A3.** Sensitivity analysis results according to the system construction cost.

| System Construction Cost | Current Worth Method | Benefit/Cost Ratio | Annual Work Cost Reduction Rate | Rate of Return | Breakeven Point |
|---|---|---|---|---|---|
| 60% ($7191.24) of the existing construction cost | $64,978.18 | 5.87 | 82.95% | 109.08% | 1.62 years |
| 80% ($9588.32) of the existing construction cost | $60,527.05 | 4.40 | 77.27% | 79.50% | 2.16 years |
| 100% ($11,985.40) of the existing construction cost | $56,075.91 | 3.52 | 71.59% | 61.48% | 2.71 years |
| 120% ($14,382.48) of the existing construction cost | $51,624.77 | 2.93 | 65.91% | 49.20% | 3.27 years |
| 140% ($16,779.56) of the existing construction cost | $47,173.63 | 2.51 | 60.22% | 40.19% | 3.83 years |
| 333.93% ($42,183.82) of the existing construction cost | $0.00 | 1.00 | 0.00% | 1.30% | 10.00 years |

**Table A4.** Sensitivity analysis results according to the system repair cost.

| System Repair Cost | Current Worth Method | Benefit/Cost Ratio | Annual Work Cost Reduction Rate | Rate of Return | Breakeven Point |
|---|---|---|---|---|---|
| 5% ($599.27) of annual repair cost | $56,075.91 | 3.52 | 71.59% | 61.48% | 2.71 years |
| 10% ($1198.54) of annual repair cost | $50,490.31 | 2.81 | 64.46% | 56.27% | 3.41 years |
| 15% ($1797.81) of annual repair cost | $44,904.72 | 2.34 | 57.33% | 51.00% | 4.11 years |
| 20% ($2397.08) of annual repair cost | $39,319.12 | 2.01 | 50.20% | 45.66% | 4.82 years |

**Table A5.** Sensitivity analysis results according to the discount rate.

| Discount Rate | Current Worth Method | Benefit/Cost Ratio | Annual Work Cost Reduction Rate | Rate of Return | Breakeven Point |
|---|---|---|---|---|---|
| 0.52% (minimum inflation rate) | $58,984.48 | 3.60 | 72.21% | 61.48% | 2.73 years |
| 0.9% (minimum bank deposit interest rate) | $57,545.07 | 3.56 | 71.91% | 61.48% | 2.72 years |
| 1.3% (reference) | $56,075.91 | 3.52 | 71.59% | 61.48% | 2.71 years |
| 1.88% (average bank deposit interest rate) | $54,025.87 | 3.46 | 71.12% | 61.48% | 2.70 years |
| 3.12% (maximum inflation rate) | $49,940.04 | 3.34 | 70.09% | 61.48% | 2.69 years |
| 3.43% (maximum bank deposit interest rate) | $48,977.70 | 3.31 | 69.83% | 61.48% | 2.68 years |

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
