# Peer review of "2D-LiDAR-Sensor-Based Retaining Wall Displacement Measurement System"

_applsci, doi:10.3390/app122211335_

Round 1

Reviewer 1 Report

General comment

The proposed work is quite interesting and I learn a lot from it. The following changes and clarifications are needed before it is recommended for publication.

Comment 1

In the abstract (lines 16-17 of the manuscript), the authors stated that the proposed method could reach a root-mean-square error of 2.82 mm. It is suggested to report the measuring distance so that this level of accuracy (2.82 mm error) can be justified.

Comment 2

The authors stated that one of the challenges of using a conventional inclinometer for measuring the lateral displacement of the retaining wall is the “measurement by manpower” (line 107 of the manuscript). It seems to me that this is a matter of data collection. I am not sure if it should be considered as the limitation of the “inclinometer” itself or the limitation due to data transmission. For the method proposed in the present study (i.e., 2D LiDAR sensor), this sensor itself also cannot transmit data wirelessly. If an inclinometer is installed with a certain wireless transmission system (e.g., WiFi router), the problem of “measurement by manpower” can be reduced as well. Is it a fair comparison to justify this as the limitation of the inclinometer?

Comment 3

May I know the differences between LiDAR and the (terrestrial) laser scanner? For me, they seem like the same technology but are named differently. It appears to me that they both generate point cloud with laser illumination (see references [12, 16] of the manuscript). This clarification is necessary because the authors repeatedly argued that the TLS is expensive (e.g., lines 55-56, lines 152-153 of the manuscript). If these technologies are indeed the same/similar thing, the authors should consider revising the argument.

Comment 4

How could the deformation of a retaining wall be experimentally “simulated” by using wooden plates with different thicknesses along the height? I am not sure how to link the operation in Fig. 2 to the experiments performed in Fig. 9. Is it an out-of-plane or in-plane deformation? Certain clarifications are needed.

Comment 5

In Eq. (1), d represents the “distance data” (is it directly measured by the LiDAR?). The same symbol is adopted in line 281 and Table 3 of the manuscript for the M3C2 algorithm. Do they refer to the same physical distance (please correct me if I misunderstood it)?

Comment 6

It would be better to clarify the meaning of “Ground Truth” in Eq. (2). In Fig. 9, it seems that the authors measured only the thickness of the wood plates. How many measurement points are adopted for Eq. (2)? For me, the thickness of wood plates is like a 1D measurement. How can the reported accuracy be able to justify the accuracy in 2D?

Comment 7

I feel like better to present the hypothetical testing site in Table 4 as a figure.

Comment 8

The cost analysis is quite interesting. I wonder if the cost for learning point cloud data should be included in the discussion (although it is difficult to be quantified). I feel that using an inclinometer for measurement is more direct to the users. The point cloud data is actually not that easy to comprehend, especially since it relies on certain algorithms to construct the deformation.  

Comment 9

There are some yellow bars shown in Fig. 16. May I know what is the difference between the yellow ones and the blue ones? Are they the “reference” shown in Appendix B?

Comment 10

In line 278, there is a citation error. The authors use brackets to cite reference [30], thus, the year “(2013)” is not needed.

Comment 11

Will the rotation speed affect the accuracy? If it is too fast, I believe that it will induce additional errors in the measurement.

Comment 12

In Fig. 10, there is a term called “Reference point-cloud data”. Is it the initial measurement taken by the LiDAR?

Author Response

The authors sincerely appreciate the reviewer’s thoughtful comments and suggestions on our manuscript. Each of the reviewer’s comments and suggestions was carefully reviewed and considered while revising the manuscript. Please check the attached file for the point-by-point response according to the reviewer's comment.

Reviewer 2 Report

The problems when measuring retaining-wall displacements using inclinometers are as follows; a) difficulties installing them, b) identifying the 2D local lateral displacement of the retaining wall, and c) measuring using manpower. In order to overcome these problems, the beneficial characteristics of 2D LiDAR sensor-based retaining-wall displacement measurement system were carefully addressed with a plenty of relevant past reports and were verified by reliable experimental approaches. The performance of the designed system was developed practically, and evaluated by the suitable LCC analysis; its reliability regarding the measurement results and economic feasibility of the system were well demonstrated.

Author Response

Thank you for your considerate and constructive comments on this manuscript. 
